# Quantitative effects of environmental variation on stomatal anatomy and gas exchange in a grass model

Tiago D. G. Nunes[1] , Magdalena W. Slawinska[1] , Heike Lindner[1] and Michael T. Raissig[1,2]

[1]Centre for Organismal Studies Heidelberg, Heidelberg University, Heidelberg, Germany; [2]Institute of Plant Sciences, University of Bern, Bern, Switzerland

## Original Research Article

*Brachypodium distachyon,* environmental influence; grass stomata; leaf-level gas exchange; stomatal size and density.

**Address correspondence to:**
Michael T. Raissig
E-mail: michael.raissig@ips.unibe.ch

## Abstract

Stomata are cellular pores on the leaf epidermis that allow plants to regulate carbon assimilation and water loss. Stomata integrate environmental signals to regulate pore apertures and adapt gas exchange to fluctuating conditions. Here, we quantified intraspecific plasticity of stomatal gas exchange and anatomy in response to seasonal variation in *Brachypodium distachyon*. Over the course of 2 years, we (a) used infrared gas analysis to assess light response kinetics of 120 Bd21-3 wild-type individuals in an environmentally fluctuating greenhouse and (b) microscopically determined the seasonal variability of stomatal anatomy in a subset of these plants. We observed systemic environmental effects on gas exchange measurements and remarkable intraspecific plasticity of stomatal anatomical traits. To reliably link anatomical variation to gas exchange, we adjusted anatomical $g_s$max calculations for grass stomatal morphology. We propose that systemic effects and variability in stomatal anatomy should be accounted for in long-term gas exchange studies.

## 1. Introduction

Stomata are the cellular pores on the leaf epidermis that allow plants to balance photosynthetic carbon dioxide ($CO_2$) uptake with water vapor loss. Stomatal movements result from changes in turgor of stomatal cells (Jezek & Blatt, 2017). Stomatal opening is induced by an increase of turgor pressure in guard cells (GCs), while a decrease of turgor pressure in GCs results in stomatal closure. To optimise gas exchange, stomata interpret and integrate a plethora of environmental cues such as light, humidity, temperature, $CO_2$ concentration and even biotic factors like pathogens (Engineer et al., 2016; Jezek & Blatt, 2017; Kollist et al., 2014; Merilo et al., 2014; Murata et al., 2015; Sierla et al., 2016). In high light, for example, stomata of $C_3$ and $C_4$ plants open to provide sufficient $CO_2$ for photosynthesis. In low light, on the other hand, less $CO_2$ is required to saturate photosynthesis and, consequently, stomata close to limit water loss. Therefore, stomatal responsiveness and fast opening and closing kinetics can significantly contribute to plant water use efficiency (WUE) in changing environments (Lawson & Vialet-Chabrand, 2019; McAusland et al., 2016). WUE represents the ratio of carbon assimilation and water loss and is a crucial trait for plant productivity and stress resilience (Leakey et al., 2019; McAusland et al., 2016). Grasses, which include the cereals like rice, maize and wheat, show comparatively fast stomatal movements that likely contribute to more water-efficient gas exchange in changing environments (Franks & Farquhar, 2007; Lawson & Matthews, 2020; McAusland et al., 2016).

During the day plants face changing environmental conditions such as fluctuating ambient light intensity ($Q_{out}$), temperature ($T$) and relative humidity (RH). Stomata mostly respond locally to environmental stimuli. This allows infrared gas analyser (IRGA)-based leaf gas exchange studies to be robust since leaves are placed in a chamber and exposed to controlled $Q_{out}$, RH, $T$ and $CO_2$ concentration ($[CO_2]$) regardless of the ambient conditions. Nevertheless, it has already been suggested that external ambient conditions might systemically affect local stomatal responses measured by IRGA systems (Devireddy et al., 2018; Devireddy et al., 2020;

Ehonen et al., 2020). This might be particularly relevant for gas exchange studies that are performed in greenhouse or field settings with significant daily and seasonal environmental fluctuations. However, the putative systemic influence of the varying ambient conditions on gas exchange measurements is not generally accounted for.

Furthermore, gas exchange parameters such as carbon assimilation ($A$), stomatal conductance to water vapor ($g_{sw}$), intrinsic water use efficiency (iWUE) and stomatal kinetics are influenced by anatomical traits such as stomatal density (SD) and stomatal length (SL) (Elliott-Kingston et al., 2016; Faralli et al., 2019; Haworth et al., 2021; Lawson & Blatt, 2014). SD and SL are negatively correlated and vary in response to a variety of environmental conditions such as $T$, RH, [$CO_2$] or $Q_{out}$ (Bertolino et al., 2019; Franks et al., 2009; Zhang et al., 2021). The seasonal variation of environmental conditions might, therefore, affect the intraspecific plasticity of stomatal anatomical traits influencing gas exchange performance and, consequently, the results of long-term gas exchange phenotyping studies.

Here, we quantified stomatal conductance kinetics in 120 individuals of the grass model *Brachypodium distachyon* (Bd21-3) in a greenhouse over the course of 2 years. Simultaneously, we logged the environmental conditions in the greenhouse ($Q_{out}$, $T$ and RH) and time of the day (time) and quantified how these parameters affected the measured gas exchange parameters ($A$, $g_{sw}$, iWUE and stomatal response kinetics). We additionally quantified anatomical traits of stomata (SD and SL) in three different seasons (summer, autumn and winter) and observed a significant impact of seasonal growth conditions on these traits. This allowed us to correlate how variations in SD and SL influence steady-state gas exchange, stomatal kinetics and maximum stomatal conductance ($g_s$max). When calculating anatomical $g_s$max based on anatomical traits, we realised that existing approaches to calculate maximum pore area for the double end-correction version of the equation by Franks and Farquhar (2001) did not sufficiently account for the graminoid morphology. Using quantitative morphometry of open and closed *B. distachyon* stomata we determined how to accurately calculate maximum pore area and pore depth. These adjustments allowed for an accurate prediction of physiological $g_s$max based on stomatal anatomical traits in *B. distachyon*.

## 2. Material and methods

### 2.1. Plant material and growth conditions

*B. distachyon* Bd21-3 seeds were vernalised in water for 2 days at 4°C before being transferred to soil. Plants were grown in a greenhouse with 18 hr light:6 hr dark, average day temperature = 28°C, average night temperature = 25°C and average RH = 40%. We used $6 \times 6 \times 8$ cm pots per plant filled with four parts soil (Einheitserde CL ED73) and one part vermiculite. The greenhouse is located at 49° 24' 52.38" N and 8° 40' 5.808" E at the Centre for Organismal Studies Heidelberg, Im Neuenheimer Feld 360, 69120 Heidelberg, Germany. Daily mean temperature (January 2019 to September 2021) varied between 3 and 5°C in December to February, 8–11°C in March to April, 13–21°C in May to September and 6–12°C in October to November (Deutscher Wetterdienst, https://cdc.dwd.de/portal/). Average daylight hours are 8.3–10.2 hr in December to February, 11.9–13.8 hr in March to April, 12.6–16.2 hr in May to September and 9.1–10-8 hr in October to November (Deutscher Wetterdienst, https://cdc.dwd.de/portal/).

### 2.2. Leaf-level gas exchange measurements

All measurements were performed on *B. distachyon* leaves 3 weeks after sowing using a LI-6800 Portable Photosynthesis System (Li-COR Biosciences Inc., Lincoln, NE, USA) equipped with a Multiphase Flash Fluorometer (6800-01A) chamber. The youngest, fully expanded leaf was measured using the 2 $cm^2$ leaf chamber. Conditions in the LI-6800 chamber for light-response experiments were as follows: flow rate, 500 $\mu$mol/s; fan speed, 10,000 rpm; leaf temperature, 28°C; RH, 40%; [$CO_2$], 400 $\mu$mol/mol; photosynthetic active radiation (PAR), 1,000–100–1,000–0 $\mu$mol PAR m$^{-2}$ s$^{-1}$ (20 min per light step) (Figure 1a). Light-response measurements of $A$ and $g_{sw}$ were obtained for 120 wild-type Bd21-3 individuals between May 2019 and September 2021. Gas exchange measurements were automatically logged every minute. Relative $g_{sw}$ was calculated by normalising $g_{sw}$ to the highest $g_{sw}$ value observed to evaluate kinetics of stomatal response regardless of variation on absolute $g_{sw}$, eliminating the influence of stomatal density and leaf area. Because *B. distachyon* leaves do not fill the 2 $cm^2$ chamber, individual leaf area was measured for a subset of 35 individuals to accurately quantify absolute $g_{sw}$ and $A$. To obtain a mean approximation of gas exchange levels for the total 120 individuals, absolute $g_{sw}$ and $A$ were corrected by using the average leaf area (0.64 $cm^2$) from the data subset ($n = 35$). Intrinsic WUE (iWUE) was calculated as the $A$ to $g_{sw}$ ratio ($A/g_{sw}$). Ambient light intensity ($Q_{out}$) was monitored during the measurements using an external LI-190R PAR Sensor (Li-COR Biosciences Inc., Lincoln, NE, USA) attached to LI-6800. Greenhouse temperature and RH were monitored during the experiments using an Onset HOBO U12-O12 4-channel data logger (Onset Computer Corporation, Bourne, MA, USA) that was placed next to the plants used for analysis. One-phase decay or one phase association non-linear regressions were obtained for the stomatal closure transitions (1,000–100 and 1,000–0 PAR) and stomatal opening transition (100–1,000 PAR), respectively, to determine half-time ($T_{50\%}$) and rate constant ($k$). Maximum stomatal conductance ($g_s$max) measurements were performed with the following conditions: flow rate, 500 $\mu$mol/s; fan speed, 10,000 rpm; leaf temperature, 28°C; RH, 68–70%; [$CO_2$], 100 $\mu$mol/mol; PAR, 1,500 $\mu$mol PAR m$^{-2}$ s$^{-1}$. Gas exchange measurements were automatically logged every minute and $g_s$max was calculated as the average of the last 5 min at steady-state.

### 2.3. Microscopy analysis of stomatal anatomical traits

Leaves of a subset of individuals ($n = 4–6$ per season; $n = 5$ in summer, $n = 6$ in autumn, $n = 4$ in winter) were collected after LI-6800 measurements and fixed in 7:1 ethanol:acetic acid. To prepare samples for imaging, leaf tissue was rinsed in water and mounted on slides in Hoyer's solution. The abaxial side was imaged using a Leica DM5000B (Leica Microsystems, Wetzlar, Germany). For SD, 3–5 fields of view (0.290 $mm^2$, 20× objective) per leaf were counted resulting in 60–160 stomata per individual. For SL and width ($W_A$), stomata from 4 to 6 fields of view (0.0725 $mm^2$, 40× objective) per leaf were measured resulting in 20–70 stomata per leaf.

### 2.4. Correlation analysis and statistics

Correlation analysis for gas exchange parameters independent of leaf area like iWUE (ratio between $A$ and $g_{sw}$) and half-time of opening/closing were performed for all 120 individuals. Correlation analysis for gas exchange parameters dependent on leaf area like $g_{sw}$ and $A$ were only performed for the subset of 35 individuals, for which the accurate individual leaf area was determined.

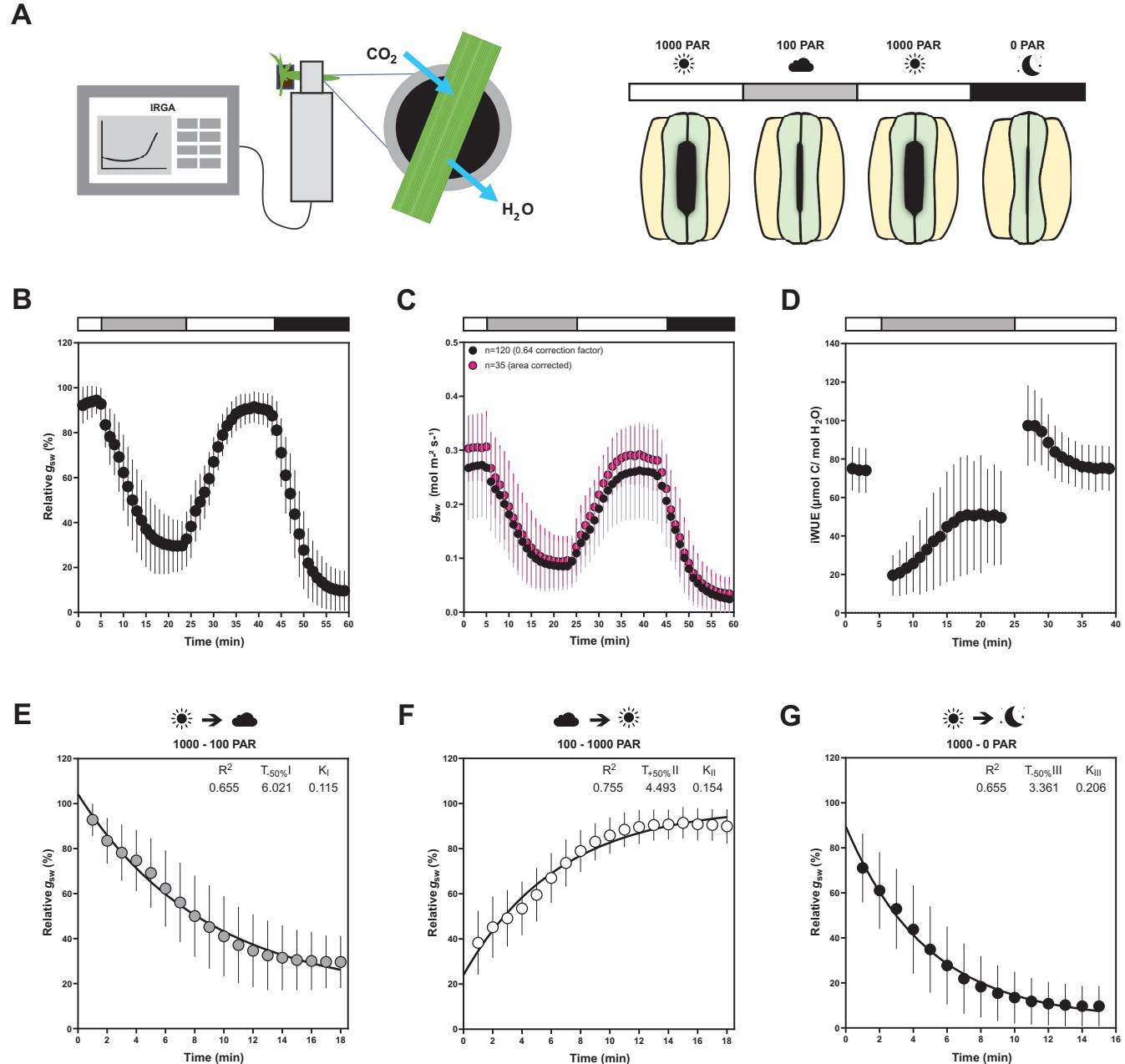

**Fig. 1.** Leaf-level gas exchange measurements in response to changing light intensities reveal fast and consistent stomatal movements in *Brachypodium distachyon* Bd21-3. (a) Experimental setup for measuring leaf-level gas exchange parameters related with $CO_2$ capture and $H_2O$ vapor loss by clamping a leaf in an infrared gas analyser (IRGA) chamber with controlled environmental conditions. Gas exchange is measured in changing light conditions (1,000–100–1,000–0 PAR) inducing stomatal closure in response to decreasing light intensity/darkness and stomatal opening in response to increasing light intensity. (b) Relative stomatal conductance (Rel $g_{sw}$) during the light transitions ($n = 120$, normalised to highest $g_{sw}$ observed). (c) Absolute stomatal conductance ($g_{sw}$) response to light transitions (in black, data from 120 individuals corrected by average leaf area of 0.64 $cm^2$ and in magenta data from a subset of 35 individuals corrected by individual leaf area). (d) Intrinsic water-use efficiency (iWUE) response to light transitions (1,000–100–1,000 PAR) ($n = 120$, calculated as $A/g_{sw}$). (e) One-phase decay exponential regression for the transition 1,000–100 PAR ($n = 120$). (f) One-phase association exponential regression for the transition 100–1,000 PAR ($n = 120$). (g) One-phase decay exponential regression for the transition 1,000–0 PAR ($n = 120$). $R^2$, half-time ($T_{\pm50\%}$) and rate constant ($K$) are indicated. Error bars = SD.

For the correlation analysis between steady-state gas exchange and environmental conditions, the last 5 min of steady-state gas exchange parameters ($A$, $g_{sw}$ and iWUE) at the second, third and fourth light step (100, 1,000 and 0 PAR) and their corresponding ambient conditions ($Q_{out}$, RH and $T$) were averaged. Pearson correlation matrices represented as heatmaps were obtained for steady-state $g_{sw}$, $A$, iWUE, $Q_{out}$, RH, $T$ and time at 100, 1,000 and 0 PAR. ROUT method was used to remove outliers (Motulsky & Brown, 2006).

For the correlation analysis between stomatal kinetics and steady-state $g_{sw}$, Pearson correlation matrices represented as heatmaps were obtained for steady-state initial/final $g_{sw}$ and half-time ($n = 35$). For the correlation between stomatal kinetics and environmental conditions, Pearson correlation matrices represented as heatmaps were obtained for half-time, initial/final $T$, RH $Q_{out}$, and time of the day ($n = 120$).

Relevant correlations between different pairs of parameters were represented with linear or non-linear regressions. Significant

($p$ < .05) and non-significant linear regressions were represented with solid and dashed lines, respectively. Non-linear regressions (quadratic function) were chosen for correlations with time of the day as we observe an axis of symmetry around noon, for which a non-linear model was biologically more appropriate.

For the correlation analysis between stomatal anatomical traits and growth environmental conditions, Pearson correlation matrices represented as heatmaps were obtained for SD, SL and environmental growth conditions (average $T$, average RH and day length). Pearson correlation matrices and linear regressions were obtained for the correlation analysis between stomatal anatomy and gas exchange parameters (steady-state $g_{sw}$, A, iWUE and $T_{50\%}$).

To test for significant differences between two groups we performed an unpaired $t$-test. One-way ANOVA followed by Tukey's multiple comparison test was used when comparing more than two groups. $p$ values are indicated directly in the graphs. All analyses were performed on GraphPad Prism version 9.1.0, GraphPad Software, San Diego, CA, USA, www.graphpad.com.

### 2.5. Stomatal morphometric analysis and anatomical $g_s$max calculations

To characterise fully open stomata, leaves were treated with 4 $\mu$M fusicoccin (Santa Cruz Biotechnology, Inc., Dallas, TX, USA) solution in opening-closing buffer (50 mM KCl and 10 mM MES-KOH). Collected leaves were dipped into 70% ethanol and infiltrated with fusicoccin solution. For infiltration, a needleless syringe was used to infiltrate the leaf tissue on the adaxial side until the tissue was visibly wet. Infiltrated leaves were then cut into smaller pieces (approx. 3–5 mm long) and incubated overnight in fusicoccin solution in the light. To analyse closed stomata, leaves were treated with 50 $\mu$M ABA (Merck, Darmstadt, Germany) solution in opening-closing buffer (50 mM KCl and 10 mM MES-KOH) as described for fusicoccin treatment and incubated overnight in the dark. Before imaging on the confocal microscope, leaves were stained with propidium iodide (1%) (Thermo Fisher Scientific, Waltham, MA, USA). Z-stacks of 30 open and 30 closed stomata from 3 different individuals each were taken on the Leica TCS SP8 confocal microscope (Leica Microsystems, Wetzlar, Germany). The obtained z-stacks were analysed using Fiji. For each stoma, pore length (PL), pore width (PW) at the centre of the pore, guard cell length (GCL), right and left guard cell width at the middle of the stoma (GCW$_C$) and stoma width at the apices were measured ($W_A$) on the z-sum projection image. To measure the exact pore area, each pore was manually traced with the polygon selection tool. To measure pore depth ($l$), the central pore part was selected with the rectangle selection tool and resliced starting at the top, avoiding interpolation. Pore depth was measured on the z-sum projection of the reslice.

The leaves assessed for physiological $g_s$max were fixed with ethanol:acetic acid 7:1. GCL, $W_A$ and stomatal width at the centre ($W_C$) were measured on light microscope pictures (20–40 stomata per leaf). GC width was calculated as half of $W_A$ or $W_C$. SD was obtained by counting the number of stomata in five different areas per leaf (20× objective).

For the anatomical maximum stomatal conductance calculations, we used the anatomical $g_s$max equation from Franks and Farquhar (2001) (see Figure 5h), where SD is the stomatal density (stomata mm$^{-2}$), $a_{max}$ is the maximum pore area ($\mu$m$^2$), $l$ is the pore depth ($\mu$m), $d$ is the diffusivity of water in air (0.0000249 m$^2$/s$^{-1}$, at 25°C), $\nu$ is the molar volume of air (0.024464 m$^3$/mol$^{-1}$, at 25°C)

and $\pi$ is the mathematical constant. Maximum pore area was either measured by hand-tracing the stomatal pore of fully open stomata or calculated as an ellipse (with major axis equal to pore length and minor to half the pore length) or a rectangle (pore width × pore length).

## 3. Results

### 3.1. B. distachyon *shows fast and consistent stomatal gas exchange in response to changing light*

The plants' physiology including stomatal gas exchange dynamics are strongly influenced by the environmental conditions the plant is exposed to (Arve et al., 2013; Durand et al., 2020; Matthews et al., 2017). Closed-system IRGA allow gas exchange measurements within a chamber with tightly controlled environmental settings regardless of ambient conditions (Douthe et al., 2018). However, plants have previously developed in and acclimated to a specific environment. Furthermore, during measurements, most distal parts of the plant remain exposed to ambient environmental conditions that might significantly differ from the conditions in the IRGA chamber (Figure 1a). To quantify the consistency of stomatal responses and the influence of variable greenhouse conditions on gas exchange, we analysed gas exchange parameters and kinetics of 120 wild-type *B. distachyon* individuals (Bd21-3) over the course of 2 years in a partially environmentally controlled greenhouse. The ambient conditions in the greenhouse varied remarkably over the course of the 120 IRGA measurements (Supplementary Figure S1A–C) and the measurements covered a broad range of hours of the day (time) from 6 am to 7 pm (Supplementary Figure S1D). We obtained consistent and reproducible stomatal light-responses ($R^2$ = 0.66–0.76) (Figure 1b,e–g) despite variation in absolute stomatal conductance ($g_{sw}$) levels (Figure 1c). Because the *B. distachyon* leaf is smaller than the 2 cm$^2$ chamber used, $g_{sw}$ was corrected using the average leaf area from a data subset for which we measured and corrected for the actual individual leaf area ($n$ = 35) (Supplementary Figure S1E), to obtain a mean approximation of gas exchange levels for the total 120 individuals. The data subset corrected with the actual leaf area (Figure 1c, magenta dots) nicely overlapped with the average correction of the 120 individuals (Figure 1c, black dots) and together revealed reasonable variation of absolute $g_{sw}$ values. Importantly, the 35 leaf area corrected measurements covered the range of environmental conditions observed in all 120 individuals (Supplementary Figure S1A–D).

The first light transition (1,000–100 PAR) resulted in a 70% decrease in $g_{sw}$ with a half-time of 6 min ($T_{-50\%}I$ = 6.021 min) (Figure 1e). An increase in light intensity (100–1,000 PAR) induced an exponential increase in $g_{sw}$ with a half-time of less than 5 min ($T_{+50\%}II$ = 4.493 min) until reaching similar $g_{sw}$ as in the previous high light step (1,000 PAR) (Figure 1f). Switching from 1,000 to 0 PAR resulted in strikingly fast stomatal closure with a half-time of only ~3 min ($T_{-50\%}III$ = 3.361 min) and, thus, represented the quickest of the three light transition responses (Figure 1g). $g_{sw}$ was on average 0.29 ± 0.06 mol m$^{-2}$ s$^{-1}$ at high light and 0.10 ± 0.05 mol m$^{-2}$ s$^{-1}$ at low light (Figure 1c). We observed an average of 0.015 ± 0.013 mol m$^{-2}$ s$^{-1}$ of residual $g_{sw}$ in darkness (Figure 1c). At high light, iWUE was on average 77 ± 13 $\mu$mol CO$_2$/mol H$_2$O, whereas at low light iWUE was 51 ± 26 $\mu$mol CO$_2$/mol H$_2$O (Figure 1d). A was on average 21 ± 4 $\mu$mol m$^{-2}$ s$^{-1}$ at high light and 4 ± 1 $\mu$mol m$^{-2}$ s$^{-1}$ at low light (Supplementary Figure S1F).

Together, *B. distachyon* shows fast stomatal light responses typical for grasses, which were consistent over 120 measurements.

## 3.2. Quantitative effects of greenhouse environmental fluctuations on gas exchange in B. distachyon

To quantify how the different environmental conditions affected gas exchange, we performed correlation analysis between gas exchange parameters [stomatal conductance ($g_{sw}$), carbon assimilation ($A$) and intrinsic water-use efficiency (iWUE)] and environmental conditions [temperature ($T$), ambient light intensity ($Q_{out}$), RH, time of the day (time)]. Correlations were done separately for low light steady-state (100 PAR, Figure 2a,g), high light steady-state (1,000 PAR, Figure 2d,h), steady-state in darkness (0 PAR, Supplementary Figure S2E) and for opening and closing kinetics (Supplementary Figure S2H–K). Because exact leaf area was only measured for a subset of 35 individuals to calculate accurate absolute $g_{sw}$ and $A$, the correlation analysis between environmental parameters and absolute gas exchange parameters (i.e., $g_{sw}$ and $A$) was performed using the 35 individuals only. On the other hand, the 120 samples were used for correlation analysis between environmental conditions and leaf-area independent parameters like iWUE and stomatal kinetics (half-time).

iWUE was negatively correlated with $T$ at both light conditions 100 and 1,000 PAR (Figure 2a,d). Increasing temperatures significantly associated with decreasing iWUE values (Figure 2b,e). iWUE also correlated with time in both light conditions (Figure 2a,d). A quadratic relation can be observed between time and iWUE (Figure 2c,f), particularly at low light, with the lowest iWUE reached at midday (Figure 2c). Similar correlations between iWUE and $T$ or time were observed in the data subset ($n$ = 35) (Figure 2g,h).

Regarding the steady-state gas exchange parameters, $T$ showed a considerable influence on $g_{sw}$ at both low and high light (Figure 2g,h) and $g_{sw}$ significantly increased with rising ambient temperatures (Figure 2i,j). $Q_{out}$, on the other hand, correlated with both $A$ and $g_{sw}$ at high light (Figure 2h). Both $A$ and $g_{sw}$ significantly increased with increasing $Q_{out}$ (Figure 2k,l). Together, this explained why iWUE is only correlated with $T$ but not with $Q_{out}$.

Lastly, time significantly correlated with $g_{sw}$ at all light conditions and also with $A$ at low light (Figure 2g,h and Supplementary Figure S2A–E,G). No significant correlations were observed between ambient conditions ($Q_{out}$, $T$ or RH) and $g_{sw}$ at 0 PAR (Supplementary Figure S2E–G), even though an influence of $T$ on $g_{sw}$ is suggested (Supplementary Figure S2F) as observed at 100 and 1,000 PAR (Figure 2i,j).

Finally, stomatal kinetics (i.e., half-time $T_{50\%}$) significantly depended on the initial and/or final steady-state $g_{sw}$ (Supplementary Figure S2H), which in turn were affected by the environment (see above). In addition, our data also suggested an influence of diurnal rhythm (time) on stomatal closure speed (half-time $T_{50\%}$) as stomata seem to close slower at noon (Figure 2m–o and Supplementary Figure S2I–K).

In summary, fluctuations in ambient conditions such as temperature and light intensity during measurements, and diurnal rhythm influenced steady-state gas exchange parameters and/or stomatal kinetics within a strictly controlled IRGA leaf chamber, which highlights the relevance of considering systemic effects on stomatal physiology experiments.

## 3.3. Seasonal changes in greenhouse growth conditions affect stomatal anatomical traits and gas exchange

While the artificial light intensity and light-darkness cycles are controlled, the contribution of ambient sunlight intensity and day length (DL), average temperature ($T$) and RH vary among seasons in our greenhouse (Supplementary Figure S3A,D–F). Stomatal anatomical traits such as SD and stomatal size are strongly influenced by environmental cues to which the plants are exposed to during development (Casson & Gray, 2008; Liu et al., 2018; Qi & Torii, 2018; Terfa et al., 2020). To test the plasticity and variability of stomatal anatomical traits in *B. distachyon* wild-type plants, we quantified SD and SL as a proxy for stomatal size from 15 individuals grown in different seasons – summer (May to June, $n$ = 5), autumn (October to November, $n$ = 6) and winter (January to February, $n$ = 4) – and correlated these traits with growth conditions ($T$, RH and DL; Figure 3a and Supplementary Figure S3A). In summer, SD was ~40% higher than in winter (105.8 ± 13.1 vs. 74.2 ± 6.8 stomata per mm$^2$) and SL reduced by ~10% (24.6 ± 0.7 vs. 26.8 ± 0.3 $\mu$m) (Supplementary Figure S3A–C). Consequently and as previously described (Franks & Beerling, 2009; Haworth et al., 2021; Zhang et al., 2021), we observed a strong negative correlation between SD and SL (Figure 3a,d) and a strong correlation between anatomy and environment (Figure 3a). SD and SL correlated in an opposite manner with the different environmental parameters (Figure 3a). An increase in $T$ correlated with an increase in SD and a decrease in SL (Figure 3b,c). Since $T$ and RH were negatively correlated, RH correlated in an opposite manner with SD and SL (Figure 3e,f). SD and SL were also inversely correlated with DL, with longer days associated with shorter stomata and higher SD (Supplementary Figure S3G,H). Overall, summer plants grown during longer days with higher ambient light intensity, higher $T$, and lower RH (Supplementary Figure S3D–F), developed higher SD and lower SL (Supplementary Figure S3B,C). In autumn and winter, plants grown during shorter days with lower ambient light intensity, lower $T$ and higher RH (Supplementary Figure S3D–F), developed lower SD and higher SL (Supplementary Figure S3B,C). However, which environmental parameter primarily caused changes to stomatal anatomy is unclear.

Besides a seasonal variation on stomatal anatomical traits, we observed seasonal variation on gas exchange (Figure 3g–i). Our data suggested slightly, yet not significantly higher $A$ and lower $g_{sw}$ in summer under high light compared to autumn and winter (Figure 3g,h). Yet, iWUE was significantly higher in summer than in autumn and winter under high light conditions (Figure 3i). Our data also suggests an increase in $A$ and $g_{sw}$ from autumn to winter (Figure 3g,h). Under low light conditions (100 PAR), no differences in $A$ occurred among seasons (Figure 3g). On the other hand, higher $g_{sw}$ was observed in autumn and winter (Figure 3h), resulting in lower iWUE at 100 PAR (Figure 3i). When measuring physiological maximum stomatal conductance ($g_s$max) of autumn/winter plants, the anatomical offset between SD and SL seemed to compensate for stomatal gas exchange maximum capacity, even though a non-significant decrease in average $g_s$max was observed in autumn/winter (Supplementary Figure S3I). In conclusion, stomatal anatomical traits of wild-type *B. distachyon* are surprisingly plastic and variable among seasons even in a semi-controlled growth environment, likely contributing to the seasonal variation on functional traits.

## 3.4. Stomatal anatomical traits influence gas exchange

Due to the seasonal variation in stomatal anatomy and functional traits, we quantified how the anatomical variation translates into changes in functional traits such as steady-state gas exchange parameters (at high light 1,000 PAR, low light 100 PAR and darkness 0 PAR) and stomatal kinetics (Supplementary Figure S4).

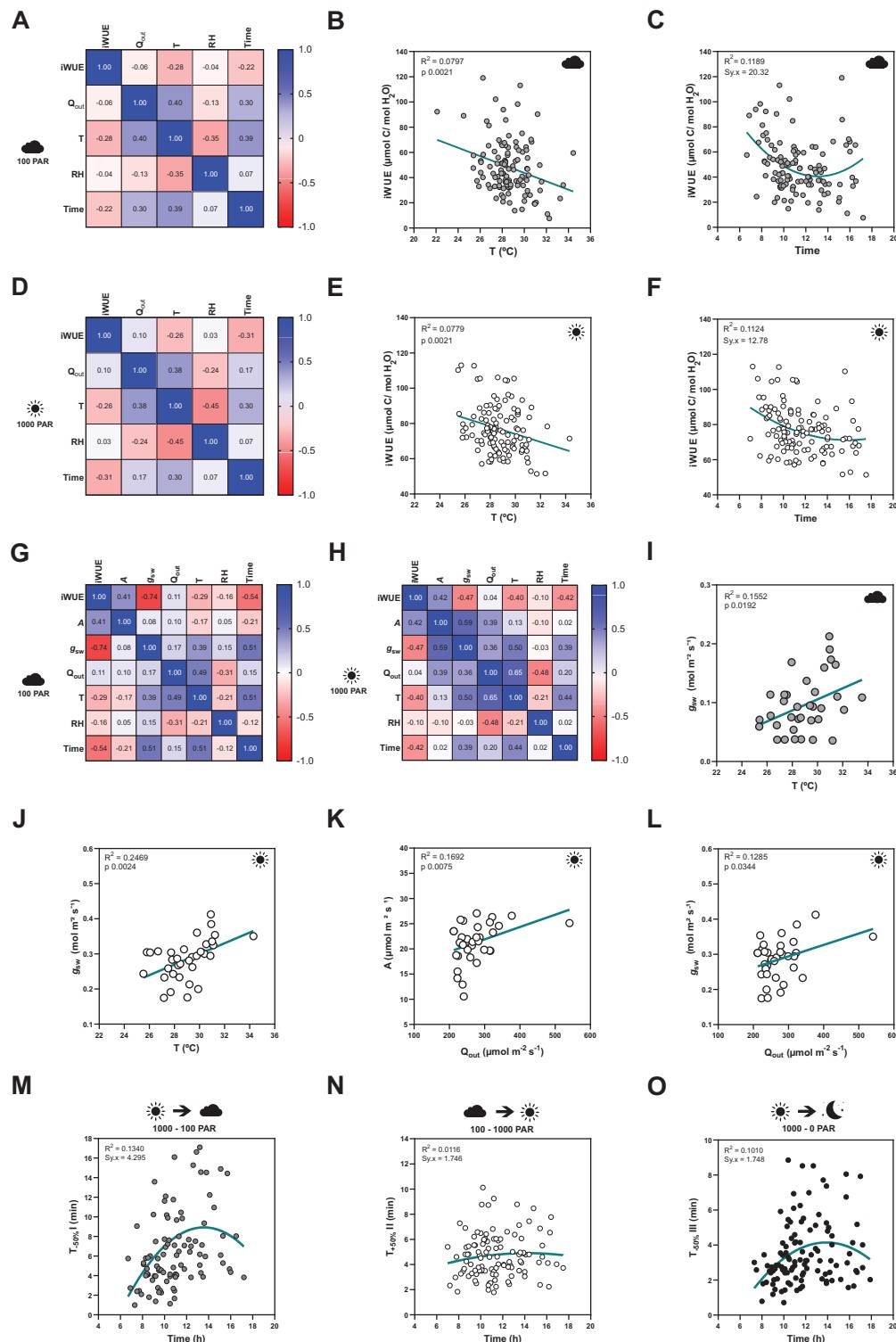

**Fig. 2.** Influence of temperature, ambient light and diurnal rhythm on leaf-level gas exchange. (*A*) Correlation matrix between iWUE and environment ($Q_{out}$, $T$, RH and time) of 120 measurements of wild-type *B. distachyon* (Bd21-3) at 100 PAR (second light step). (b) Linear regression between $T$ and iWUE at 100 PAR ($n$ = 116). (c) Non-linear regression between time and iWUE at 100 PAR ($n$ = 116). (d) Correlation matrix between iWUE and environment ($Q_{out}$, $T$, RH and time) of 120 measurements of wild-type *B. distachyon* (Bd21-3) at 1,000 PAR (third light step). (e) Linear regression between $T$ and iWUE at 1,000 PAR ($n$ = 119). (f) Non-linear regression between time and iWUE at 1,000 PAR ($n$ = 120). (g) Correlation matrix between gas exchange parameters (*A*, $g_{sw}$ and iWUE) and environment ($Q_{out}$, $T$, RH and time) of the 35 measurements (corrected by individual leaf area) of wild-type *B. distachyon* (Bd21-3) at 100 PAR (second light step). (h) Correlation matrix between gas exchange parameters (*A*, $g_{sw}$ and iWUE) and environment ($Q_{out}$, $T$, RH and time) of the 35 measurements (corrected by individual leaf area) of wild-type *B. distachyon* (Bd21-3) at 1,000 PAR (third light step). (i) Linear regression between $T$ and $g_{sw}$ at 100 PAR ($n$ = 35). (j) Linear regression between $T$ and $g_{sw}$ at 1,000 PAR ($n$ = 35). (k) Linear regression between $Q_{out}$ and *A* at 1,000 PAR ($n$ = 35). (l) Linear regression between $Q_{out}$ and $g_{sw}$ at 1,000 PAR ($n$ = 35). (m) Non-linear regression between half-time of the transition 1,000–100 PAR ($T_{-50\%}$I) and time of the day (time) ($n$ = 99). (n) Non-linear regression between half-time of the transition 100–1,000 PAR ($T_{+50\%}$II) and time of the day (time) ($n$ = 111). (o) Non-linear regression between half-time of the transition 1,000–0 PAR ($T_{-50\%}$III) and time of the day (time) ($n$ = 111). $R^2$ and Sy.x or $p$ values are indicated.

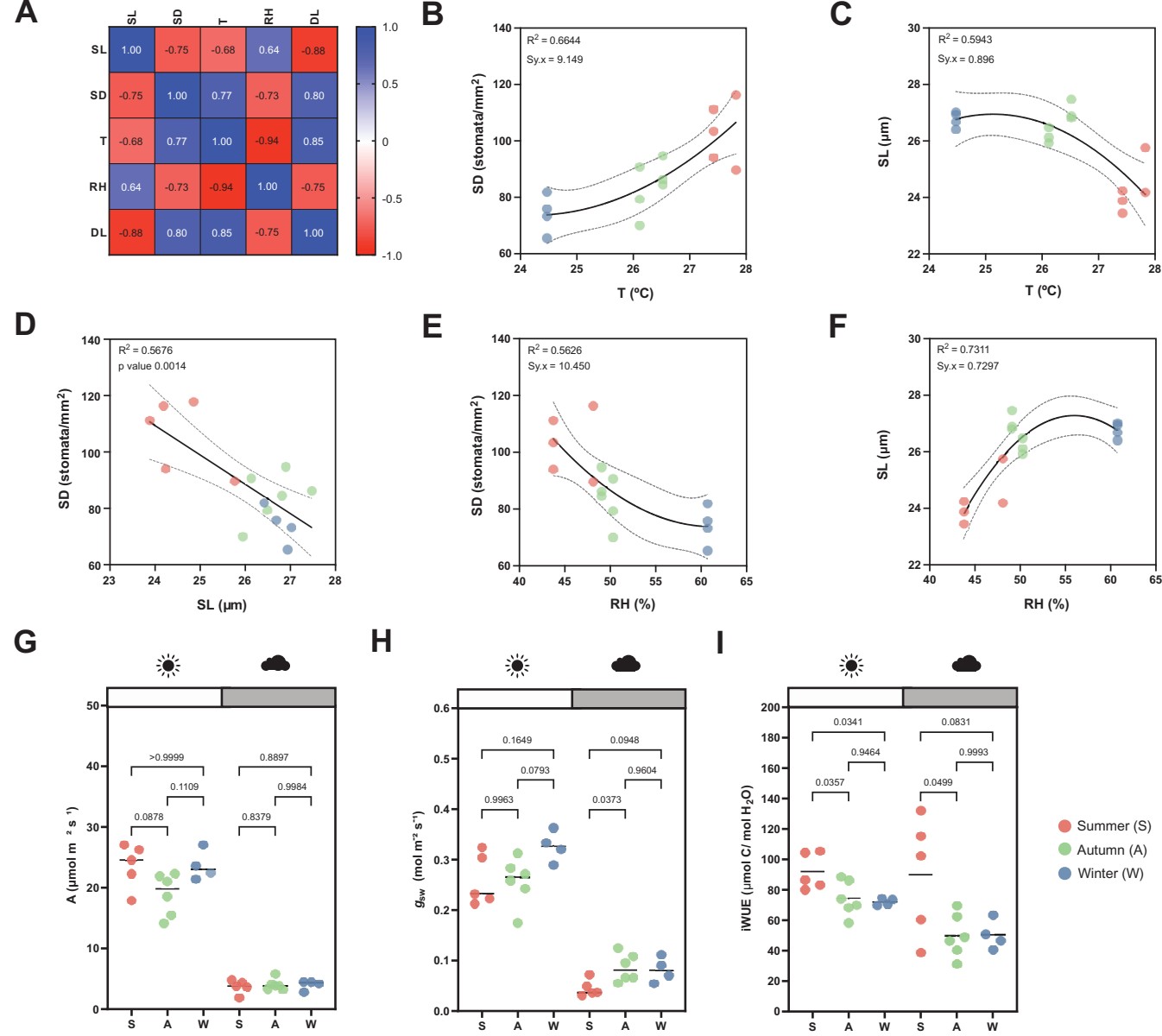

**Fig. 3.** Effects of seasonal growth conditions on stomatal anatomical traits and on gas exchange. (a) Correlation matrix between stomatal length (SL), stomatal density (SD), average growth temperature ($T$), average growth relative humidity (RH) and day length (DL) ($n = 15$). (b) Non-linear quadratic (second order polynomial regression) relation between $T$ and SD ($n = 15$). (c) Non-linear quadratic (second order polynomial regression) relation between $T$ and SL ($n = 15$). (d) Linear relation between SD and SL ($n = 15$). (e) Non-linear quadratic (second order polynomial regression) relation between RH and SD ($n = 15$). (f) Non-linear quadratic (second order polynomial regression) relation between RH and SL ($n = 15$). (g) Seasonal variation on $A$ at 1,000 (white) and 100 (grey) PAR ($n = 4$–$6$ per season; $n = 5$ in summer, $n = 6$ in autumn, $n = 4$ in winter). (h) Seasonal variation on $g_{sw}$ at 1,000 and 100 PAR ($n = 4$–$6$ per season; $n = 5$ in summer, $n = 6$ in autumn, $n = 4$ in winter). (i) Seasonal variation on iWUE at 1,000 and 100 PAR ($n = 4$–$6$ per season; $n = 5$ in summer, $n = 6$ in autumn, $n = 4$ in winter). $R^2$ and Sy.x or $p$ values are indicated. Dashed lines in (B-F) indicate 95% confidence bands.

In terms of steady-state $g_{sw}$, lower SD (and higher SL, to a lesser extent) are the anatomical traits associated with higher operational stomatal conductance ($g_{sw}$) in *B. distachyon* (Figure 4a,b). Similarly, an increase in SD and a decrease in SL resulted in an increase of $A$ at high light, while no effect was observed in 100 PAR (light limiting condition) (Figure 4c,d). Consequently, higher SD and lower SL result in higher iWUE (Figure 4e,f), even though SD had a stronger effect on iWUE (Figure 4f). Thus, the higher iWUE observed in summer (Figure 3i) might be primarily caused by the higher SD and lower SL observed in this season (Supplementary Figure S3B,C). The correlations between anatomical traits (SD and SL) and $g_{sw}$ were stronger in low light than in high light (Supplementary Figure S4 and Figure 4a,b) likely contributing to the

higher seasonal variation in iWUE at low light than in high light (Figure 3i).

Regarding stomatal kinetics ($T_{50\%}$), the effect of SL on stomatal closure and opening was non-significant (Figure 4g and Supplementary Figure S4). However, while the influence of SD on stomatal opening was also weak (Figure 4h and Supplementary Figure S4), surprisingly stronger correlations and significant effects were observed between SD and stomatal closure kinetics ($T_{-50\%}$) (Figure 4h and Supplementary Figure S4). High SD was strongly correlated with water-use efficiency as the increase of SD led to higher steady-state iWUE (Figure 4f) and faster stomatal closure (Figure 4h) contributing to higher water-use efficiency in changing environments. Overall, stomatal anatomy is strongly correlated

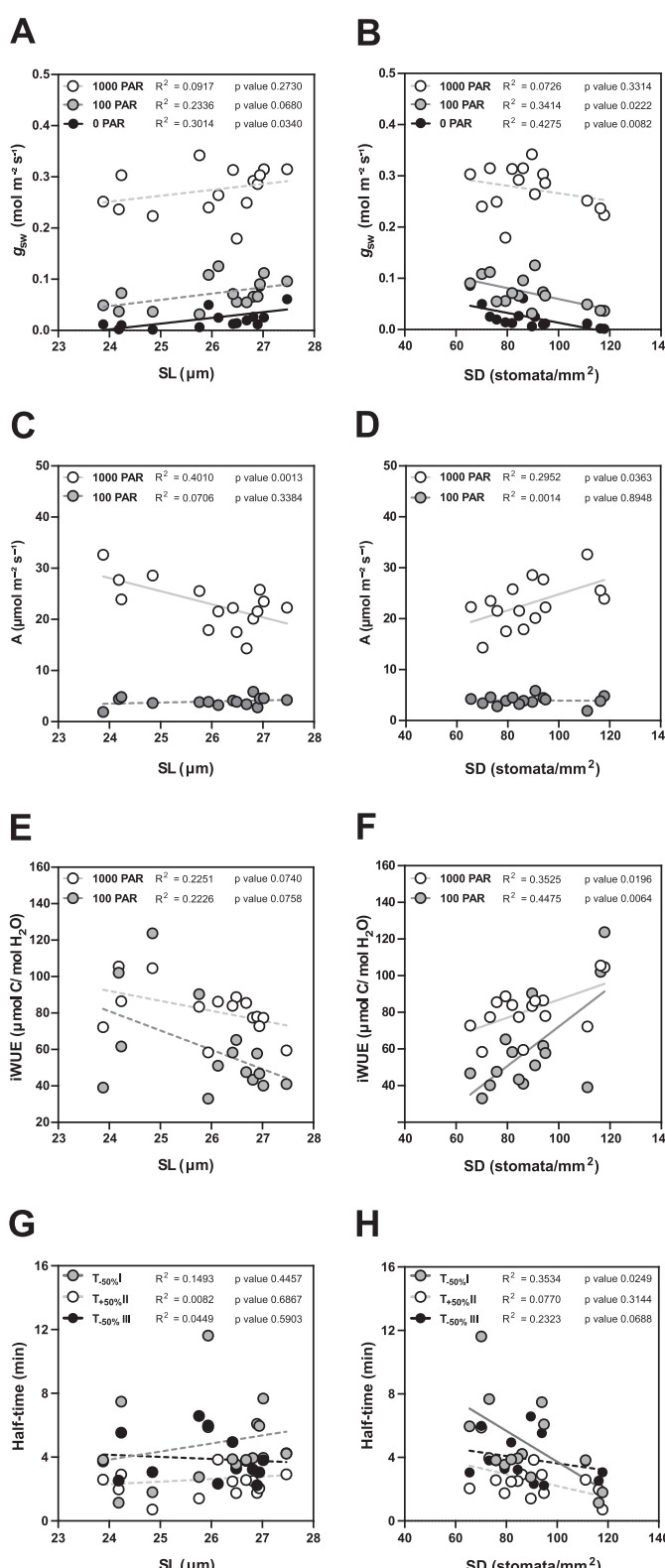

**Fig. 4.** Impact of stomatal anatomical traits on steady-state gas exchange and stomatal kinetics. (a) Linear regressions between stomatal length (SL) and $g_{sw}$ at 1,000 (white dots), 100 (grey dots) and 0 (black dots) PAR ($n = 15$). (b) Linear regressions between stomatal density (SD) and $g_{sw}$ at 1,000 (white dots), 100 (grey dots) and 0 (black dots) PAR ($n = 15$). (c) Linear regressions between SL and $A$ at 1,000 (white dots) and 100 PAR (grey dots). (d) Linear regression between SD and $A$ at 1,000 (white dots) and 100 PAR (grey dots). (e) Linear regressions between SL and iWUE at 1,000 PAR (white dots) and 100 PAR (grey dots) ($n = 15$). (f) Linear regressions between SD and iWUE at 1,000 PAR (white dots) and 100 PAR (grey dots) ($n = 15$). (g) Linear regressions between SL and half-time ($T_{50\%}$) of the light transitions 1,000–100 (grey dots), 100–1,000 (white dots) and 1,000–0 (black dots) PAR ($n = 15$). (h) Linear regressions between SD and $T_{50\%}$ of the light transitions 1,000–100 (grey dots), 100–1,000 (white dots) and 1,000–0 (black dots) PAR ($n = 15$). $R^2$ and $p$ values are indicated. Dashed lines indicate statistically non-significant linear regressions ($p > 0.05$).

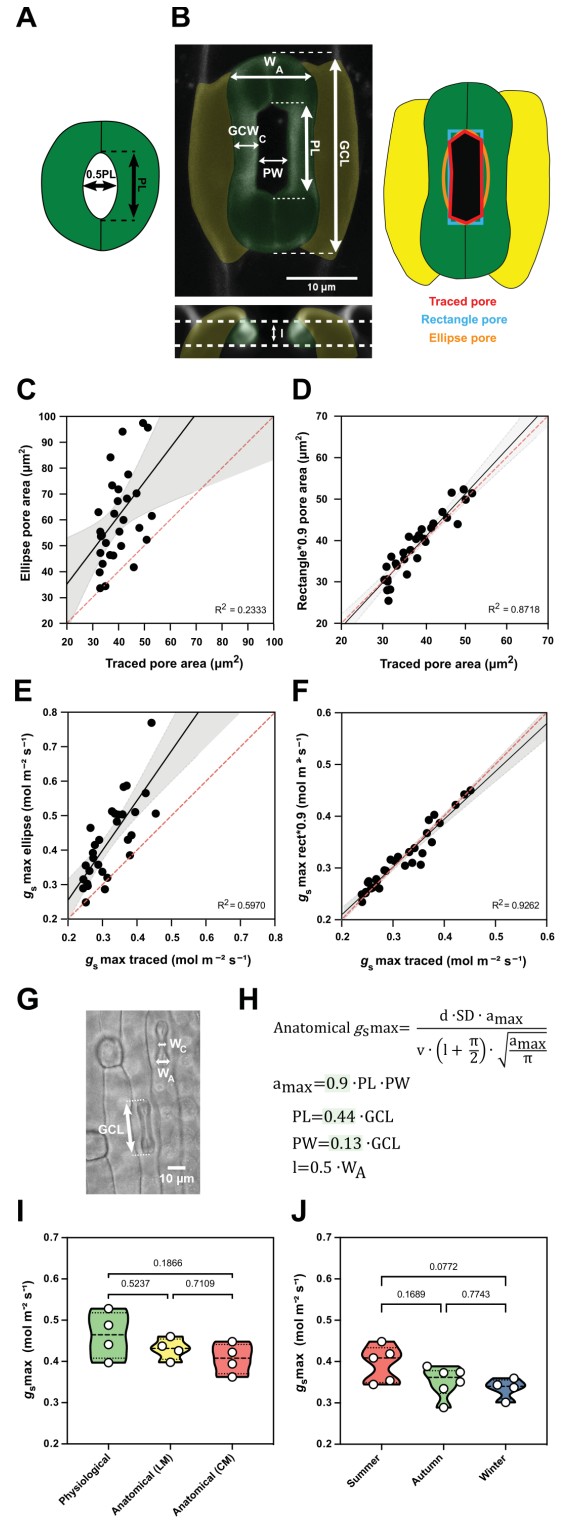

**Fig. 5.** Morphometric analysis of graminoid *B. distachyon* stomata significantly improves anatomical $g_s$max predictions. (a) *Arabidopsis*-like stoma and ellipse pore shape. (b) *B. distachyon* stomatal morphology traits measured; guard cell length (GCL), pore length (PL), pore width (PW), guard-cell width at the centre of the stomata (GCW$_C$), stomatal width at the apex ($W_A$) and pore depth ($l$). Pore area hand-traced (red) or geometrically defined as an ellipse (orange) or a rectangle (blue). (c) Linear relation between hand-traced pore area and ellipse pore area. (d) Linear relation between hand-traced pore area and rectangle multiplied by 0.9. (e) Linear relation of anatomical $g_s$max calculated with hand-traced pore area and with ellipse pore. (f) Linear relation of anatomical $g_s$max calculated with hand-traced pore area and with rectangle pore multiplied by 0.9. (g) Anatomical parameters measured using light microscopy; stomatal width at the apex ($W_A$) and GCL. (h) Anatomical maximum stomatal conductance (anatomical $g_s$max) equation as defined by Franks and Farquhar (2001) and *B. distachyon* adjustments to calculate $a_{max}$ (0.9∗PL∗PW), PL (0.44∗GCL), PW (0.13∗GCL) and $l$ ($W_A$∗0.5). (i) Comparison between physiological $g_s$max, anatomical $g_s$max (light microscopy, LM) and anatomical $g_s$max (confocal microscopy, CM). (j) Comparison of anatomical $g_s$max calculated for summer, autumn and winter plants with stomatal anatomical traits represented in Supplementary Figure S3B,C. $R^2$ and $p$ values are indicated.

ductance ($g_s$max) by calculating the theoretical anatomical $g_s$max, which is based on the anatomical traits SD, maximum pore area ($a_{max}$) and pore depth ($l$). While SD is assessed for any species simply by counting stomata per leaf area, formulae to calculate maximum pore area ($a_{max}$, $\mu m^2$) and pore depth ($l$, $\mu m$) were optimised for *Arabidopsis*-like stomatal morphologies and ellipsoid pores (Dow et al., 2014; Franks & Farquhar, 2001) (Figure 5a).

High-resolution confocal stacks of fusicoccin (fus)-treated open stomatal complexes in *B. distachyon* revealed hexagonal rather than elliptical pores (Figure 5b). We performed careful morphometric analysis of open and closed stomata to characterise GCL, PL, PW, GCW$_C$ and stomatal $W_A$ (Figure 5b). Furthermore, we manually traced and measured pore areas of fus-treated complexes to empirically determine a more appropriate way to calculate $a_{max}$ of graminoid stomata. Calculating $a_{max}$ as for *Arabidopsis*-like stomata (ellipse with major axis equal to pore length and minor to half the pore length, Figure 5a) caused significant pore area overestimation compared to the manually traced pores of fus-treated open complexes (Figure 5b,c). A rectangular rather than an ellipsoid approach still overestimated pore area (Figure 5b and Supplementary Figure S5A), as the manually measured pore areas approximated 90 ± 7% of the rectangular pore area (Supplementary Figure S5C). Thus, *B. distachyon* stomatal pore area can be accurately estimated using rectangular pore area calculations multiplied by a correction factor of 0.9 (Figure 5d).

Regarding pore depth ($l$), we observed that the mean pore depth measured from orthogonal resliced confocal stacks (3.37 ± 0.4 $\mu m$) approximates the mean GC width at the centre (3.23 ± 0.3 $\mu m$) but not the GC $W_A$ (5.92 ± 0.4 $\mu m$) in fully opened stomata (fus-treated) (Supplementary Figure S5G). In closed stomata (ABA-treated), on the other hand, the mean GC width at the centre was 2.48 ± 0.3 $\mu m$ while the GC $W_A$ was 3.52 ± 0.6 $\mu m$ (Supplementary Figure S5G). Therefore, if not exactly measured from orthogonal sections, then pore depth could be approximated as central GC width of fully open stomata or apical GC width of closed stomata.

We then calculated anatomical $g_s$max using hand-traced $a_{max}$ and estimated $a_{max}$ using formulae for (a) ellipse, (b) rectangle pore and (c) rectangle pore multiplied by the correction factor 0.9. Pore depth was measured from orthogonal resliced confocal stacks (Figure 5b) and stomatal density was determined by counting stomata in 3–5 different fields of view using light microscopy. We could observe that anatomical $g_s$max using hand-traced $a_{max}$ nicely correlated with anatomical $g_s$max calculated for the rectangle pore multiplied by the correction factor 0.9 (Figure 5f). This was not the

with stomatal functioning and the seasonal variation in stomatal anatomy strongly contributed to seasonal variation in gas exchange.

### 3.5. Morphometric analysis of graminoid stomata to optimise anatomical $g_s$max predictions in *B. distachyon*

Finally, we wanted to mathematically describe the impact of the observed trade-off between SD and SL on maximum stomatal con-

case when using ellipse $a_{max}$ (Figure 5e) or rectangular pore without the correction factor (Supplementary Figure S5B,C).

To determine $a_{max}$ from simple light microscopy pictures, where pores are hard to see, we calculated correction coefficients to estimate pore length and width from GC length. By calculating the ratios of the morphometrically determined GCL, PL and PW, we found that PL is 44 ± 3% of GCL and PW is 13 ± 3% of GCL (Supplementary Figure S5D–F). Thus, for calculations using light microscopy pictures, we estimated PL as 0.44∗GCL and PW as 0.13∗GCL (Figure 5g,h). To approximate pore depth ($l$), GC $W_A$ was used for closed or partially open stomata (½ of the stoma $W_A$) (Figure 5g,h and Supplementary Figure S5G) and GC width at the centre was used for fully open stomata.

Next, we tested if our adjustments for anatomical $g_s$max calculations could be used to reliably predict physiological $g_s$max in *B. distachyon*. We performed IRGA-based measurements of $g_s$max (physiological $g_s$max) in four independent individuals, collected these exact leaf zones, and measured anatomical traits from segments of those by using both standard light microscopy (after fixation with ethanol:acetic acid 7:1) and confocal microscopy (after treatment with fusicoccin). No significant differences were found between physiological $g_s$max and anatomical $g_s$max based on measured anatomical parameters using light microscopy (LM) or confocal microscopy (CM) (Figure 5i and Supplementary Figure S5H). In summary, the optimised formula for accurate anatomical $g_s$max estimation can be used to reliably predict physiological $g_s$max in *B. distachyon*.

Finally, we calculated anatomical $g_s$max for the summer, autumn and winter individuals whose anatomical traits (SD and SL) were shown in Supplementary Figure S3B,C. Even though a decrease in average anatomical $g_s$max was observed in autumn and winter, this difference was non-significant (Figure 5j). These results match our observations in physiological $g_s$max measurements between summer and autumn/winter plants (Supplementary Figure S3I). Ultimately, these findings suggest that the trade-off between SD and SL in wild-type *B. distachyon* might serve as a mechanism to maximise stomatal conductance capacity in different environments.

## 4. Discussion

Consistent and reproducible stomatal kinetics were observed for *B. distachyon* regardless of the variable greenhouse conditions. *B. distachyon* displayed the fast stomatal movements typical for grass species, which are faster than most non-grass species with kidney-shaped GC (Franks & Farquhar, 2007; Grantz & Assmann, 1991; McAusland et al., 2016; Merilo et al., 2014). The quick stomatal movements of grasses like *B. distachyon* are associated with the graminoid morphology, where two lateral subsidiary cells (SCs) flank the central, dumbbell-shaped GCs (Gray et al., 2020; Nunes et al., 2020; Stebbins & Shah, 1960). Fast stomatal movements in grasses require SCs (Raissig et al., 2017), which might function as specialised ion reservoirs (Raschke & Fellows, 1971) and mechanically accommodate GC movement to accelerate both stomatal opening and closing (Franks & Farquhar, 2007). In addition, the reduced volume-to-surface ratio of dumbbell-shaped GCs likely requires less exchange of water and ions to be pressurised (Franks & Farquhar, 2007).

Furthermore, no major asymmetry between closure and opening speed was observed. This is consistent with the results from a comparison of stomatal kinetics between eight species with kidney-shaped GCs and seven species with dumbbell-shaped GCs, where species with dumbbell-shaped GCs displayed the quickest responses and showed the most similarity between opening and closure times (McAusland et al., 2016). Nonetheless, the fastest stomatal movement in *B. distachyon* was dark-induced stomatal closure (1,000–0 PAR). Faster stomatal closing than opening has been previously described for several species and suggested to be a water-conserving strategy (Lawson & Vialet-Chabrand, 2019; Leakey et al., 2019; McAusland et al., 2016). Fast stomatal movements are important to quickly adjust stomatal pores to avoid excess of water vapor loss through stomata ($g_{sw}$) during suboptimal carbon assimilation ($A$) in low light. Intrinsic water use efficiency (iWUE, the ratio between $A$ and $g_{sw}$), varied between 50 and 80 $\mu$mol/mol which is consistent with the range of iWUE described for other $C_3$ grass species such as wheat (25–65 $\mu$mol/mol) and rice (50–80 $\mu$mol/mol) (Giuliani et al., 2013; Jahan et al., 2014).

Despite the consistent stomatal responsiveness in *B. distachyon*, we observed a significant influence of the time of the day on light-response stomatal kinetics. While diurnal variation on gas exchange has been well described for $C_3$ species (de Dios, 2017; Matthews et al., 2017; Miao et al., 2021; Roussel et al., 2009; Stangl et al., 2019; Vahisalu et al., 2008), the observed diurnal variation on stomatal responsiveness to changing light in the tightly regulated conditions of the IRGA chambers was compelling. Stomatal closure and opening speed were mainly affected by the time of the day and by steady-state $g_{sw}$ prior and/or after the change in light intensity. A similar dependence of $g_{sw}$ kinetics on light intensity transitions, on the time of the day and on steady-state $g_{sw}$ prior to light intensity changes has been recently described in *Musa spp.* (Eyland et al., 2021).

Steady-state $g_{sw}$, on the other hand, was significantly influenced by the environmental conditions. Stomatal conductance was stimulated by increasing ambient temperatures. This effect has been observed and described as a leaf cooling mechanism to cope with higher temperatures (Gommers, 2020; Lamba et al., 2018; Sonawane et al., 2021; Urban et al., 2017a; 2017b). Yet, after exceeding a certain threshold, such high temperatures may lead to stress-induced stomatal closure (Faria et al., 1996; Ikkonen et al., 2015; Yamori et al., 2006; Zhou et al., 2015). In addition, increasing ambient light intensity also significantly impacted IRGA measurements by triggering increases in $g_{sw}$ and $A$ levels. Recent studies have also reported systemic stomatal responses to light and heat in Arabidopsis (Devireddy et al., 2018; 2020) and in response to darkness and elevated $CO_2$ in birch and poplar (Ehonen et al., 2020). Consistently, our results suggest that *B. distachyon* stomata integrate both local environmental cues and systemic signals from distal parts of the plant. Therefore, it is important to monitor environmental conditions and consider their impact on gas exchange measurements, particularly in greenhouse or field studies with significant environmental fluctuations.

Apart from the effect of environmental conditions on gas exchange, different growth conditions had a major impact on anatomical traits. SD and SL inversely varied among seasons due to significant variation in ambient growth conditions. The seasonal trade-off between SD and SL mostly maintained maximum gas exchange capacity in the different growth environments. In addition, higher SD was associated with faster stomatal closure and the combination of higher SD and lower SL associated with improved iWUE in wild-type *B. distachyon*. This suggests that higher SD and lower SL, which are associated with improved stomatal responsiveness and more water use efficient gas exchange, are a morphological adaptation to summer. Higher stomatal

densities in warmer environments were reported and associated as an ecophysiological significant response for leaf evaporative cooling (Carlson et al., 2016; Hill et al., 2015). In contrast, autumn and winter seasons feature shorter days, decreased light intensity and colder temperatures, which can negatively affect photosynthesis (Feng et al., 2018; Yamasaki et al., 2002). Thus, a decrease in stomatal density to increase the leaf surface allocated to light harvesting, compensated by an increase in stomatal size to maintain maximum gas exchange capacity, might sustain optimal gas exchange and photosynthesis in winter. While we observed higher iWUE in *B. distachyon* wild-type plants with lower SL and higher SD, crop species (wheat, barley and rice) overexpressing an inhibitor of stomatal development (*EPF1*) show a reduction in SD and improved iWUE (Caine et al., 2018; Dunn et al., 2019; Hughes et al., 2017). Thus, genetically modifying SD (and/or SL) in *B. distachyon* beyond the intraspecific range of variation might allow to improve iWUE. *B. distachyon* genotypes varying in single morphological traits may help to better understand the independent influence of SL and SD on gas exchange kinetics, capacity and water-use efficiency, and verify the correlations observed in this present study.

Some studies described a negative correlation between stomatal size and stomatal speed (e.g., (Drake et al., 2013; Durand et al., 2020; Kardiman & Rbild, 2018). In a study comparing different rice genotypes, stomatal size was negatively correlated with stomatal half-time (Zhang et al., 2019), with larger stomata being faster. In a comparison of 15 different species (including 7 grass species), however, smaller stomata were associated with faster stomatal movements (McAusland et al., 2016). Other approaches using a broader range of species from different plant groups (by comparing 7 (Elliott-Kingston et al., 2016) and 31 (Haworth et al., 2018) different species) suggest that stomatal speed is not related to SL but rather positively correlated with SD, as we observed in *B. distachyon*. A major effect of SD and minor effect of SL on stomatal speed under fluctuating light has also been described in a study comparing different *Arabidopsis* genotypes (Sakoda et al., 2020). Higher SD could potentially cause proximity effects triggering stomata in close vicinity to react to local changes in a more coordinated manner. However, the effect of the variation of SL and SD on gas exchange speediness may vary among species and/or genotypes. Nonetheless, environmentally induced SD and SL variation and its impact on gas exchange must be considered during long term studies performed in greenhouse or field settings.

To facilitate correlations of anatomical stomatal traits to theoretical gas exchange maximum capacity (anatomical $g_s$max) in *B. distachyon,* we adjusted established equations (Dow et al., 2014; Franks & Farquhar, 2001) to the graminoid stomatal morphology. For grass stomata, stomatal pores are rather hexagons than ellipses and the equation presented in this study accurately predicted anatomical $g_s$max for *B. distachyon*. Therefore, it can be reliably used to predict gas exchange capacity of *B. distachyon* genotypes varying in anatomical traits. Differences between the anatomical and physiological $g_s$max might reveal impaired stomatal signalling and thus, provide a tool to identify mutant phenotypes of stomatal function. In addition, anatomical $g_s$max could help to weigh the effect of variations in single morphological traits (e.g., stomatal density, stomatal size, pore area) on $g_s$max.

In conclusion, stomatal conductance kinetics are fast and consistent in the grass model species *B. distachyon*. Nevertheless and even though stomata primarily respond to the local environment (i.e., within the IRGA chamber), ambient light intensity, temperature and time of the day can have systemic effects on gas exchange influencing results from IRGA measurements. Stomatal anatomical traits are highly plastic and environmental-responsive and, furthermore, have a major impact on gas exchange. For that reason, the effect of growth conditions on stomatal anatomical traits must be considered in leaf-level gas exchange studies.

## Acknowledgements

The authors would like to thank Michael Schilbach for managing the greenhouse and gardening support. Furthermore, the authors acknowledge all members of the lab for carefully reading and critically evaluating this manuscript.

**Financial support.** This work was supported by a DAAD Study Scholarship (to M.W.S.) and by the German Research Foundation (D.F.G.) Emmy Noether Programme Grant RA 3117/1-1 (to M.T.R.).

**Conflicts of interest.** The authors declare no potential conflicts of interest.

**Authorship contributions.** T.D.G.N. and M.T.R. designed the study and conceived the experiments. T.D.G.N. and M.W.S. performed the experiments. T.D.G.N., M.W.S., H.L. and M.T.R. analysed and interpreted the data. T.D.G.N., M.W.S., H.L. and M.T.R. wrote the manuscript.

**Data availability statement.** All gas exchange data and anatomical data used in this study can be found in Supplementary Dataset 1. Stomatal images are available upon request.

**Supplementary Materials.** To view supplementary material for this article, please visit http://doi.org/10.1017/qpb.2021.19.

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
