## [Reviewer Report]

Dear Olivier

With our great pleasure we are submitting our manuscript „Quantitative effects of environmental variation on stomatal anatomy and gas exchange in a grass model“. We are convinced that it fits very well into the scope of this journal. We are taking a highly quantitative approach to measure the effect of fluctuating environmental conditions on stomatal gas exchange and stomatal anatomical traits in the model grass Brachypodium distachyon. 

Over the course of almost two years we performed leaf-level gas exchange measurements on 100 wild-type B. distachyon Bd21-3 individuals. While initially struggling with the fact that our greenhouse is not extremely well controlled and thus affecting our measurements, we decided to actually quantify the effects of fluctuating environmental conditions on our measurements, which occur within a leaf chamber head, where the environmental parameters are perfectly controlled. To evaluate both steady-state gas exchange parameters and stomatal opening and closing speed we employed a changing light regime. While the gas exchange responses were surprisingly robust, we could clearly identify an influence of the ambient environment on local gas exchange.

Furthermore, we quantified for 15 of the 100 plants how seasonal growth conditions affect stomatal anatomy. We found that both the number and size of stomata are extremely plastic within wild-type B. distachyon. The quantitative description of the anatomical parameters, ultimately allowed us to correlate how anatomical parameters affect gas exchange physiology. One of the surprises was that a higher stomatal density positively affects stomatal closure speediness. 

Finally, we adapted a formula that allows the prediction of gas exchange capacity (gsmax) from anatomical parameters to the grass stomatal morphology. Originally, this equation was formulated to account for kidney-shaped, Arabidopsis-like stomata. We used detailed confocal stacks of open and closed stomata to accurately measure the stomatal pore and then empirically optimized the formula to calculate maximum pore area and pore depth from simple light microscopy images. These adjustments allowed us to accurately predict physiological gsmax from anatomical parameters. 

To our knowledge, the dataset presented here is the first to systematically quantify intraspecific variation of stomatal gas exchange and anatomy in a semi-controlled greenhouse setting. The careful correlation analysis will aid us and others in the field to predict, which environmental parameters to consider that could affect leaf-level gas exchange measurements. Furthermore, the empirically optimized methods to calculate maximal stomatal pore area and pore depth of grass stomata will enable a more accurate prediction of gas exchange capacity in grasses. 

We would like to thank you very much for considering our work and we are looking forward to hearing from you.

With my very best wishes,

Michael

---

## [Reviewer Report]

*Comments to Author*: The research question posed by authors – how does seasonal variation in growth conditions affect gas-exchange measurements in greenhouse settings – is interesting. The question whether and how much it matters, what other leaves experience while the measured leaf is clamped into a leaf gas exchange device is important, and the presence and extent of systemic signalling affecting plant stomatal behaviour is debated and needs further research.

The manuscript presents interesting data and offers a valuable method advancement for calculation of grass anatomical gsmax. It also opens up exciting questions for further research. However, I feel that the data that forms the basis for the first two figures and conclusions based on respective regression analyses are not strong enough. My main concern is that in gas exchange analysis, true leaf area has been measured for only 15 plants of the measured 100 individuals and for others gs and A values are estimates based on leaf area of these 15 plants. The estimate may be good enough but it may not be – and currently the manuscript does not clearly show that the approximation is valid. I have some other concerns regarding the analyses shown in the first two figures that are detailed below.

The rest of the manuscript that focuses on the relationships of anatomical stomatal parameters with growth conditions and physiology, and suggests an improved approximation for calculation of anatomical gsmax, does not have such issues and in my opinion would be an interesting addition to literature in the field.

Specific comments and questions:

p7 lines 3-7 – If I understand correctly then accurate stomatal conductance measurements with determined leaf area for the part of the leaf in the IRGA chamber have only been carried out for 15 plants and for others the area has been estimated, meaning gs is also an estimate? In this case, what were the conditions during the measurement of these 15 plants (light, temperature, RH, time of day) and do they cover the range covered by all 100 plants? If not, there may be a bias in the estimate of stomatal conductance; would be good to bring out the data for this subset in the condition graphs shown in S1D. If my understanding is correct, then the numbers on S1A do not reflect real gs, as they have been calculated with the assumption of 2 cm2 chamber leaf area for all plants? If this is the case, this data gives no information on absolute values of gs and should not be shown; same applies to A shown in S1B (if the leaf area correction is valid – that is, the 15 plants conditions reflect the conditions throughout all experiments, estimated A based on leaf area correction could be shown instead).

Figure 2 – I am confused, whether the A and gs values used in model fitting are the values assuming 2 cm2 leaf area or the estimated gs using leaf area correction? As the leaf area seems to have been much smaller than the 2 cm2 (based on the 15 plants with known area), the first approach is not justified in my opinion. If the 15 plants accurately represent the conditions and size for all plants, gs and Anet estimates could be better for looking at these relationships. If real leaf area was known only for 15 plants, would be good to test if the relationships shown in Figure 2 hold or are similar for the 15 plants dataset. I realise it will be less powerful, but at the moment the data is a mix of known and estimated values, making it hard to make strong conclusions. From p8 lines 3-4 I understand that for the subset with known leaf area (15 plants), only the regressions between temperature and light with A and iWUE were significant and R2 values were larger than when these 15 plants were combined with data from plants with inaccurate leaf area – this makes me doubt that the regression models presented in Figure 2 are accurately representing the relationships in the 100 measured plants.

Another thing I am wondering about in Figure 2 and S2 is using multiple data points for one plant in analysis (conductance/A at 1000 PAR measured twice for each plant during the experiment) – I think it can be considered pseudoreplication that artificially increases n and can potentially bias the results of analysis.

Figure 3E and interpretation on p9, line 1 - The negative relationship between RH and SD is unexpected, as a body of evidence suggests that higher RH usually stimulates stomatal development (of course, the tested RHs then are usually much higher). It has been mostly studied in dicots, so maybe Brachypodium really is different. Or alternatively – the higher light and longer days of summer (S3) could stimulate stomatal development and the high SD at lower RH of summer would be caused by light instead.

Figure 4 – for nonsignificant relationships showing the trend line does not make much sense – as the slope is not significantly different from 0, the line is not helpful and can be misleading. I also got confused about the rate constant and half time – to my understanding they can be calculated from each other, making showing regressions of SL and SD with both of them redundant.

Minor comments:

S1D referred before any other panels in S1 – would be logical to make it A then.

I could not find reference to Figure 1 panel A

---

## [Reviewer Report]

*Comments to Author*: Nunes et al., characterized light response kinetics in 100 wild-type *Brachypodium distachyon* plants grown in semi controlled greenhouse conditions for almost two years. Authors used IRGA-based system for their measurements in which single leaf of the plant is positioned to the IRGA chamber with controlled environmental conditions. Authors also monitored time of the day and growth conditions (temperature, light intensity and humidity in the greenhouse) and quantified effects of these parameters on stomatal gas-exchange. Authors find that stomatal conductance was affected by temperature and light conditions in the greenhouse whereas stomatal kinetics was affected by time of the day. Previous studies have suggested that in some dicot species conditions experienced by distal parts of the plant may modify local stomatal response during IRGA measurements. Current work suggests that systemic signaling may modulate stomatal gas-exchange dynamics also in monocots.

Authors analyzed stomatal anatomical traits and showed that in *Brachypodium* stomatal anatomy shows clear seasonal plasticity. During the summer stomatal density was higher and it was accompanied by reduced stomatal size and this was correlated with faster stomatal closure and enhanced intrinsic water use efficiency. Finally, authors performed morphometric analysis by using confocal microscopy and defined pore dimensions for *Brachypodium* stomata. Further, authors developed accurate method to measure these parameters also from light microscope images, which are more convenient for large set of samples. By using these parameters, authors optimized anatomical g<sub>s</sub>max formula for grass-type stomata, and showed that it correlates well with their experimental data.

This impressive work describes in detail seasonal behavior of stomata in wild-type *Brachypodium* and reveals how sensitive stomatal physiology and especially stomatal anatomy are to subtle changes in the growth conditions. Experiments and analyses performed are thorough, conclusions are solid, and the manuscript is generally well written.

Minor comments:

1. Currently, it is indicated that n=15 for leaves used for anatomical analyses both in introduction and material and methods. However, it seems that for each season sample number varies between 4-6 and this is correctly indicated in the figure legends. This is slightly confusing. Perhaps it would be more straightforward to indicate sample number per season similarly throughout the article.

2. Page 9. “When measuring physiological maximum stomatal conductance (g<sub>s</sub>max) of autumn/winter plants, the anatomical offset between SD and SL seemed to compensate for stomatal gas exchange maximum capacity, even though a non-significant decrease in average g<sub>s</sub>max was observed in autumn/winter (Fig. S3I).”

I am curious to hear why authors choose to pool subset of autumn and winter samples for this particular analysis? (based on Figure S3 legend, 3 autumn samples and 1 winter sample have been pooled)

3. Page 9. “an increase in SL and a decrease in SD resulted in an increase of A at high light, while no effect was observed in 100 PAR (light limiting condition) (Fig.4C-D). “

Based on the data in Figure 4C-D the trends seems opposite to what was described in the manuscript. Please correct the sentence so that it will be in line with the data.

4. Page 13. “Recent studies have also reported systemic stomatal responses to light and heat in Arabidopsis (Devireddy et al., 2018, 2020; Ehonen et al., 2020). “

Ehonen et al., 2020 detected systemic effect in birch and poplar as a response to darkness and elevated CO2 treatments but actually failed to see similar effect in *Arabidopsis*. Please correct the sentence or the references so that the content is in line with referred articles.

5. Please indicate where these experiment have been performed. Perhaps it would be also useful to shortly explain typical seasonal weather in this place. This might be helpful for the reader since seasons can be dramatically different in different parts of the world.

---

## [Reviewer Report]

*Comments to Author*: Dear Dr. Raissig,

two reviewers have now read your manuscript, and as you can see, both of them found the work novel and interesting. However, they also found a few major and several minor concerns that need to be addressed. Especially, the concern related to the low number of plants in the gas exchange analysis is critical: is there an evidence to support the idea that 15 plants is sufficient?

Looking forward to handle the revised version of your manuscript.

best wishes,

Ari Pekka Mähönen

---

## [Reviewer Report]

*Comments to Author*: I think the manuscript have been greatly improved by revisions and most of my comments have been addressed.

Description of the sample number used in seasonal analysis (Figure 3) is still in part confusing. Number of samples used for this particular experiment is low and thus, it would be important to communicate it accurately.

Authors describe seasonal variation on leaf anatomy and gas exchange by using following sample groups: summer; n=5, autumn; n=6 and winter; n=4 (Figure 3). In Figure 3 panels G, H, and I Authors compare extent of seasonal variation in gas exchange traits between each seasonal groups. In the Figure 3 legend, sample number has been marked as 15 (S+A+W). This seems unusual way to describe n in this case (Figure 3 G-I), since here seasonal variation within each group is compared to the variation within another group (S vs A, S vs W, A vs W). Please indicate n so that it accurately describes number of samples in each group.

Number of data points for 1000 PAR conditions seems to be twice as high as for 100 PAR conditions for each groups. Please explain reason for this and also indicate whether or not these 1000 PAR measurements are paired. If relevant, consider also whether the statistical test used is correct.

---

## [Reviewer Report]

*Comments to Author*: The authors have carried out additional experiments and analyses to respond to questions raised in my previous review. In my opinion, the increased sample size and adjusted analyses strengthen the conclusions presented in the manuscript. The authors have found significant systemic effects of ambient conditions on gas exchange parameters measured in environmentally-controlled leaf chamber, which is an important point to consider in leaf gas exchange analyses carried out in fluctuating environments.

I have no more major questions, just a few minor points.

In Figure 1D legend: I think it should read 1000 - 100 - 1000 PAR in the brackets

Line 227: “high light steady-state (1000 PAR, Fig 2A)” – should be low light? It says 100 PAR in Fig 2A

Line 228: “steady-state in darkness (0 PAR, Fig S2A)” – should be reference to S2E?

Line 246: “influence of T on gsw is suggested (Fig. S2G)” – should be reference to S2F?

In paragraphs in lines 250-256 the authors discuss about the effects of circadian rhythms on gas-exchange, as time of day affected gas-exchange measurements. I think the term “diurnal” would be more appropriate than “circadian” about these effects. The same also applies to Figure 2 legend.

---

## [Reviewer Report]

*Comments to Author*: Dear Michael,

both reviewers were satisfied with your response to their concerns and overall they appeared pleased with the revised manuscript. They both had, though, a few minor, additional comments that sounded reasonable to me. Please, address these issues, and send a revised version of the manuscript in due time.

Looking forward to hearing from you soon

best wishes,

Ari Pekka